# Brazilian Horses from Bahia State Are Highly Infected with *Sarcocystis bertrami*

**DOI:** 10.3390/ani12243491

**Published:** 2022-12-10

**Authors:** Caroline Marques, Bruno da Silva, Yuri Nogueira, Taynar Bezerra, Aline Tavares, Waléria Borges-Silva, Luís Gondim

**Affiliations:** School of Veterinary Medicine and Animal Science, Federal University of Bahia, Salvador 40170-110, BA, Brazil

**Keywords:** *Sarcocystis fayeri*, *Sarcocystis bertrami*, prevalence, morphologic characteristic, *cox1*, equine, horse meat, sarcocyst

## Abstract

**Simple Summary:**

*Sarcocystis bertrami* is a protozoan parasite that infects horses. Equine meat containing the parasite, when consumed raw or undercooked, is suspected to cause gastrointestinal disorders in humans. Brazil exports horse meat for the European and Asian markets; however, there is no report in the country about the occurrence of *S. bertrami*. Herein, we obtained tissues from 51 horses destined for human consumption, which were examined for *S. bertrami* using morphologic and molecular methods. The parasite was observed in 100% of the examined horses. To the best of our knowledge, this is the first confirmation of *S. bertrami* in Brazilian and South American horses. Further studies are necessary to evaluate the impact of this parasite on animal and human health.

**Abstract:**

The protozoan *Sarcocystis bertrami* (syn. *Sarcocystis fayeri*) infects horses and has dogs as definitive hosts. Herein we aimed to detect *S. bertrami* in Brazilian horses destined for human consumption and to determine the frequency of infection in the examined animals. Muscle fragments from 51 horses were collected in a slaughterhouse in Bahia State during three different seasons of the year. Samples from six tissues from each animal were prepared for macroscopic and microscopic evaluation, using tissue grinding, squash and histology. *Sarcocystis* sp. was observed in 100% of the examined horses. Selected samples were processed for transmission electron microscopy (TEM). Species identification was confirmed using a PCR targeted to the mitochondrial cytochrome c oxidase subunit 1 gene (*cox1*). Histological examination revealed sarcocysts with variable sizes and shapes, and dispersed within the muscle fibers. When observed by TEM, the sarcocyst wall was wavy and covered by an electrodense layer. The villar protrusions were digitiform and bent. To our knowledge, this study is the first morphological and molecular confirmation of *S. bertrami* in horses in Brazil and South America.

## 1. Introduction

The consumption of horse meat is common in countries of east Europe and Asia [1,2]. In the last decade, due to some problems affecting bovine meat such as the pollutant dioxin, bovine encephalopathy and foot and mouth disease, there was an increase in the search for alternative red meats, including those from horses [3].

Among the pathogens that may be found in equine meat, *Sarcocystis* spp. has been reported in horses in several countries [4,5,6,7,8,9,10,11,12]. Ingestion of raw or undercooked horse meat containing tissue cysts of *Sarcocystis* spp. has been associated with food poisoning in Japan [13]; however, lesions in humans caused by ingestion of horse meat containing *Sarcocystis* sp. has never been reported. *Sarcocystis* spp. infecting equine muscles does not seem to represent a relevant clinical disease in horses and has been associated with rare causes of equine granulomatous and eosinophilic myositis [14].

Species names of *Sarcocystis* spp. found in meat from equids have been debated in several studies [15,16,17,18,19]. *Sarcocystis bertrami* and *Sarcocystis fayeri* have been described in muscle tissues from donkeys, mules and horses [4,5,6,7,8,9,10,11]. The original classification of *S. fayeri* as a new species was mostly based on the prepatent period of sporocyst shedding by dogs and on the dimensions of excreted sporocysts [5]. Almost a decade after the classification of *S. fayeri*, a study published in 1986 on *Sarcosystis* spp. derived from mules, donkeys and horses demonstrated that morphological differences of sarcocysts derived from these three animal species were so insignificant that all parasites seemed to represent a single species [11]. In recent years, studies using morphological and genetic approaches have shown that *S. bertrami* and *S. fayeri* are the same species [17,18,19]. In the current work, the species name *S. bertrami* will be used to represent both *S. bertrami* and *S. fayeri*, and the latter will be referred to as a synonym of *S. bertrami*. 

To the authors’ knowledge, until the submission of the current data there was no published paper about *S. bertrami* in the muscle tissues of Brazilian or South American horses. Thus, the aim of this study was to examine the muscle tissues of horses obtained in Bahia, Brazil, for identification of *Sarcocystis* sp. and to determine the frequency of infection, using morphological and molecular methods. 

## 2. Materials and Methods

The study was approved by the Ethical Committee for Animal Use (CEUA) of the School of Veterinary Medicine and Animal Science at Federal University of Bahia, under the number 81/2019.

### 2.1. Equine Samples 

Muscle fragments of 51 horses were obtained in a slaughterhouse under federal inspection located in Bahia State, Brazil. All samples were collected by the first author (C.M.) in three moments between 2020 and 2021. The first sampling was conducted in January 2020, the second in October 2020 and the third in April 2021. The samples (approximately 100 g per tissue) were derived from the following equine tissues: 51 from tongue, 51 from masseter, 51 from heart, 51 from diaphragm, 40 from gluteal muscle and 31 from esophagus, totalizing 275 tissue fragments. Tissues were individually packed in plastic bags and transported in Styrofoam boxes with icepacks until arrival at the Federal University of Bahia, where tissues were stored at 4 °C until analysis. Examination of animal tissues was performed until 96 h post-collection.

### 2.2. Macroscopic Examination

All 275 tissue samples were examined by the naked eye. Ten transversal cuts were performed in each sample to detect macroscopic cysts of *Sarcocystis* spp. For examination of the esophagus, it was longitudinally sectioned and its internal and external surfaces were examined for macrocysts of the parasite.

### 2.3. Microscopical Analysis Using Fresh and Stained Tissues

Samples were examined using light microscopy using squash preparations, and also by examining the sediment derived from the juice of grinded tissues. For squash preparation, three fragments of approximately 5 mm diameter of each tissue were pressed between two microscope glass slides and observed under 40× and 100× magnifications. For examination of the sediment of meat juice, 50 g of each tissue was grinded using mortar and pestle and mixed with 5–10 mL of PBS (pH 7.2). The juice was filtered with gauze and placed in a 50 mL centrifuge tube, and its final volume was completed to 45 mL with PBS for centrifugation (600× *g* for 10 min). After centrifugation, the supernatant was discarded and the sediment was suspended in 1.5 mL of PBS (pH 7.2). Three aliquots of the sediment were analyzed using light microscopy for the visualization of parasite cysts and free bradyzoites at 400× magnification. The observed cysts were collected using a needle of 19G and transferred to 1.5 ml tubes that were DNAse and RNAse free. Free bradyzoites were concentrated using centrifugation (10 min at 1200× *g*). Cysts and bradyzoites were measured and photographed using a Nikon Cli microscope with a camera and the software NIS Elements by Nikon. 

### 2.4. Histology and Transmission Electron Microscopy

Fragments of each collected tissue were fixed in buffered formalin and embedded in paraffin, and thin sections of 5 µm were conventionally prepared for hematoxylin and eosin (HE) stain. The stained sections were examined using light microscopy for parasite *Sarcocystis* sp. 

For transmission electron microcopy, tissue cysts from tongues were fixed in 2% glutaraldehyde, 0.1 M sodium cacodylate (pH 7.4), for 2 h at room temperature and then refrigerated at 4 °C. After fixation, samples were washed in 0.1 M sodium cacodylate (pH 7.4), post-fixed in 1% osmium tetroxide, dehydrated in acetone (30, 40, 50, 70, 90 and 100%) and treated with 1% phosphotungstic and 1% uranyl acetate. Then, Polybed was used as the embedding medium, which was polymerized at 60 °C. Semi-thin sections were observed using light microscopy to locate the cysts in the processed samples. Ultrathin sections were examined using a Zeiss EM900^®^ TEM. 

### 2.5. Tissue Digestion in Acid Pepsin

Muscle fragments from eight animals from the first (*n* = 3), second (*n* = 2) and third (*n* = 3) collections were employed for acid pepsin digestion. Tissues were weighted and 50 g of tissues per animal were digested in 200 mL of acid pepsin [2.5 g de pepsin (0.7 U-FIP/mg), 10 mL HCl and distilled water to 1 L]. Tissues in acid pepsin were homogenized on a magnetic mixer for 20 min at 37 °C and passed through filters of 300, 150 and 53 µm using 50 mL tubes and centrifuged for 500× *g* for 5 min. The supernatant was discarded and the sediment was washed three times in PBS (pH 7.2). The final sediment was suspended in PBS and purified using a Sephadex^®^ G-25 column, washed in PBS and concentrated using centrifugation (1500× *g* for 5 min). Bradyzoites were measured and photographed as described in Section 2.3. Then, bradyzoites were placed in 1.5 mL DNAse- and RNAse-free tubes [20].

### 2.6. Molecular Analysis

Bradyzoites derived from the tongues of two other horses (n. 11 and 13), the sediment of grinded tongue from one horse (n. 17) and tissue cysts from the tongue of another horse (n. 20) were used for molecular analysis. DNA was extracted from the samples using a commercial kit (Easy-DNA, Invitrogen^®^, Carlsbad, CA, USA), eluted in the buffer provided by the kit and stored at 4 °C until analysis. 

PCR was targeted to the gene cytochrome c oxidase subunit 1 (*cox1*) using the primers SF1 [21] and SR9 [22], which amplify a fragment of 1060 bp. Reactions were conducted in volumes of 50 μL [2 μL of each primer (10 pmol), 25 μL of a commercial PCR mix (Master Mix, Promega, Madison, WI, USA), 2 μL of DNA and 19 μL of ultrapure water]. PCR conditions were as follows: an initial denaturation step at 95 °C for 10 min, 45 amplification cycles (95 °C for 45 s, 54 °C for 45 s, 72 °C for 1 min) and a final extension at 72 °C for 10 min [21]. Positive control consisted of cultured *Sarcocystis neurona* merozoites of the strain SN-138 [23], and ultrapure water was employed as negative control. PCR products were mixed with Syber Gold (Invitrogen, Thermo Fisher Scientific, Eugene, USA) and run on 1.5% agarose gel with a 100 bp DNA ladder. The gel was visualized and documented on a UV transilluminator with a camera.

PCR products were excised from the gel and purified using a commercial DNA purification kit (PureLink™ Quick Gel Extraction and PCR Purification, Thermo Fisher Scientific, Carlsbad, CA, USA), according to the manufacturer’s instructions. Sequencing of PCR products was performed using Sanger’s method using the BigDye Terminator v3.1 Cycle Sequencing Kit (Thermo Fisher Scientific) and an ABI 3730 DNA Analyzer, Life Technologies (Applied Biosystems, Foster City, CA, USA). Nucleotide sequences were analyzed with the aid of the software FinchTV version 1.4.0. and compared with those deposited in GenBank using nucleotide BLAST (blastn). 

## 3. Results

### 3.1. Macroscopic Evaluation

No cystic structures or macroscopic lesions were observed by the naked eye in 275 tissue fragments, which were obtained from the tongue (51 animals), masseter (51), heart (51), diaphragm (51), gluteal muscle (40) and esophagus (31).

### 3.2. Microscopic Examination

No tissue cysts could be observed on *squash* preparations of the 275 tissue fragments. In contrast, one or more tissues from each of the 51 horses presented intact tissue cysts or free bradyzoites of *Sarcocystis* sp. that resulted in 100% of infected horses. The highest frequency of infection was found in the gluteal muscle (97.50%; 39/40 horses), followed by the esophagus (90.32%; 28/31), tongue (90.19%; 46/51), diaphragm (66.70%; 34/51), masseter (17.64%; 9/51) and heart (3.92%; 2/51). 

The unstained and free sarcocysts observed using light microcopy were highly septated (Figure 1A) and possessed villar protrusions arising from their cyst walls when observed at 600× magnification (Figure 1B). Cysts in muscle tissues stained by HE presented variable sizes and shapes, as elongated, elliptical or globular sarcocysts (Figure 1C–E). In longitudinal sections stained by HE, the cysts presented the following dimensions: 300.15–1.084.79 (526.89 ± 231.00) × 36.78–260.29 (103.70 ± 59.78) µm (*n* = 15). In transversal sections stained by HE, cyst dimensions were 69.95–190.80 (125.62 ± 41.22) × 29.96–126.95 (73.08 ± 28.36) µm (*n* = 15).

Free unstained bradyzoites derived from the grinded tissues or from pepsin-digested tissues presented elongated and semilunar shapes when observed using light microscopy (Figure 1F). Their dimensions were as follows: 14.03–17.64 (16.16 ± 1.03) × 3.86–5.84 (4.88 ± 0.54) µm (*n* = 20).

A total of 22 tissue sections were examined using HE. No inflammation or any other microscopic lesions were observed around the sarcocysts, although muscle fibers were increased in size due to the presence of the parasites.

### 3.3. Transmission Electron Microscopy (TEM)

Cyst walls of selected sarcocysts observed by TEM were slightly undulated, with occasional indentations, and surrounded by a thin electrodense layer. Villar protrusions were bent, presented a digitiform appearance, were sometimes folded with variable lengths (0.97–4.09 µm) and were not seen throughout the entire cyst wall. Microtubules were observed in the granular substance, as well as along the villar protrusions (Figure 2A,B). The granular substance of the cyst wall had great variation in thickness, with dimensions of 0.65–2.63 (Figure 2A,B). The cyst wall was classified as type 11c, according to a description proposed by Dubey et al. (2015) [24].

Sarcocysts were separated and each compartment around the septum was filled with zoites; bradyzoites were elongated and possessed numerous micronemes, amylopectin granules, rhoptries and a nucleus (Figure 2C–E). The conoid was seen in the apical region of bradyzoites, depending on the angle the cyst was sectioned at (Figure 2F). The septa present varied in thickness (0.7–1.0 µm) and had the same appearance as the granular substance, although no microtubules or any other structure were noted in the granular substance of the septa. 

### 3.4. Bradyzoites in Pepsin Digested Samples

Pepsin digestion performed in a limited number of gluteal tissues resulted in the visualization of free bradyzoites (Figure 1F). For each 50 g of tissues, the concentration of bradyzoites corresponded to 3–4 × 10^7^ bradyzoites. Bradyzoites were used for molecular analysis.

### 3.5. Molecular Analysis

Molecular analysis of the parasites was performed using a conventional PCR targeted to *cox1* gen. The amplified and sequenced products (OP887040-OP887043) had between 956 and 965 bp and presented 99.25% to 99.46% identities with *S. bertrami cox1* from China (GenBank: KY399759). The *cox1* sequences reported here also matched with sequences derived from Japan (LC171850; LC171840; LC171857; LC171856), although the identities were lower when compared with the Chinese one.

## 4. Discussion

The taxonomy of *Sarcocystis* sp. infecting horses has been confusing. Based on recent morphological and genetic studies, horses were demonstrated to be infected by a single species of the genus, *S. bertrami*, which has *S. fayeri* as its junior synonym [17,18]. Despite the worldwide distribution of *S. bertrami* in horses, until the conclusion of the current study, no genetic or morphological reports were found on *S. bertrami* in Brazilian or South American horses. Herein, we report *S. bertrami* infection in 100% of 51 Brazilian horses, whose meat cuts are exported for human consumption in European and Asian countries.

In Brazil, horses are not bred for the meat market, as Brazilians do not traditionally consume horse meat. Animals destined for slaughtering consist mostly of old horses. Therefore, the high frequency of *S. bertrami* infection in the tested animals may be in part due to the advanced age of the animals, leading to a higher chance of exposure to *Sarcocystis* sp. sporocysts in the environment. In addition, six different horse tissues were microscopically examined using the sediment generated by grinding horse tissues; this technique has a high sensitivity for detecting tissue cysts or bradyzoites of *Sarcocystis* spp. [25]. 

Another factor that might have contributed to the high *S. bertrami* infection rates observed in our study is related to the increasing number of abandoned donkeys in Bahia State; this animal species has been shown to also harbor *S. bertrami* in their tissues [19]. In the last century, Brazilian donkeys had a major role as working animals [26]. In recent decades, donkeys have been replaced by motorized vehicles and a large population of donkeys has been abandoned [27]. Stray donkeys are frequently found in rural areas of Bahia State and are often accidentally killed on the roads, as well as die due to poor health conditions [27]. Dogs, which are definitive hosts of *S. bertrami* [28], have an increased offer of donkey meat and may shed the parasite’s sporocysts/oocysts, favoring the transmission of *S. bertrami* to horses. It is unknown whether other wild Brazilian canids, besides dogs, act as definitive hosts of *S. bertrami*, although Brazilian domestic dogs have also not been tested as definitive hosts of *S. bertrami* in Brazil.

The frequency of *S. bertrami* infection in horse tissues worldwide is variable, depending, in part, on the kind of examined tissues and the employed diagnostic techniques. In a report from China, *S. bertrami* was detected in 73.9% (34/46) of the horses using squash examination of five different tissues [18]; these authors examined 40 squash preparations from each tissue. In our study, we have found 100% of 51 horses infected with *S. bertrami*, whose tissues were examined by grinding 50 g of each of six different tissues. Our analysis using squash preparation yielded negative results; however, we have used only three squash preparations for each kind of horse tissue, contrasting with the number of squash preparations (*n* = 40) in the Chinese study [18]. 

It is worth noting that the only published Brazilian study that aimed to detect *Sarcocystis* sp. in heart tissues from 197 horses had negative results [29]. In a report from China, heart samples of 46 horses were tested for *Sarcocystis* sp. and the parasite could not be found in any examined heart [18]; in contrast, these authors detected 73.9% (34/46) of *Sarcocystis* sp.-infected animals when other tissues (esophagus, diaphragm, skeletal muscle and tongue) from the same 46 horses were examined for the parasite. Based on our results and in previous reports [18,29], *S. bertrami* has a low tropism for equine heart tissues, whereas gluteal muscle, esophagus, diaphragm and tongue are more often infected by the parasite.

Sediment of grinded tissues contained free bradyzoites or intact sarcocysts. Free bradyzoites with approximate lengths between 14 and 17 µm were classified as *Sarcocystis* sp. because bradyzoites and tachyzoites from related parasites (*Toxoplasma gondii* and *Neospora caninum*) have lengths shorter than 9 µm [30]. Identification of *S. bertrami* was confirmed by the ultrastructural characteristics of the cyst wall observed by TEM, as well as by PCR and sequencing of the mitochondrial gene *cox1*. Tissue cysts of selected samples presented cyst walls classified as type 11c [24], which were similar to previous descriptions of *S. bertrami* in other continents. Interestingly, the dimensions of bradyzoites, villar protrusions and the granular substance were very close to the dimensions reported in China [18]. The sequence of *cox1* had a *coverage* of 96% and identity of 99.46% with a *cox1* Chinese sequence (KY399759) of *S. bertrami* [17]. 

Future studies are necessary to evaluate whether *S. bertrami* observed in Brazilian horses represent any risk to human health, and to assess potential association with the parasite and muscle alterations in horses. Additional studies should also focus on the use of other molecular markers, besides the cox1 gene, to explore potential molecular differences among *S. bertrmi* found in Brazil and isolates of the parasites in other countries. 

## 5. Conclusions

Brazilian horses from Bahia State were highly infected with *S. bertrami*. The high percentage of the parasite in gluteal muscle (97.5%), which represents an appreciated meat cut consumed by humans, calls attention to the putative risk of ingesting the parasite tissue cysts. To the best of our knowledge, this study represents the first molecular and morphological confirmation of *S. bertrami* in Brazil and South America.

## Figures and Tables

**Figure 1 animals-12-03491-f001:**
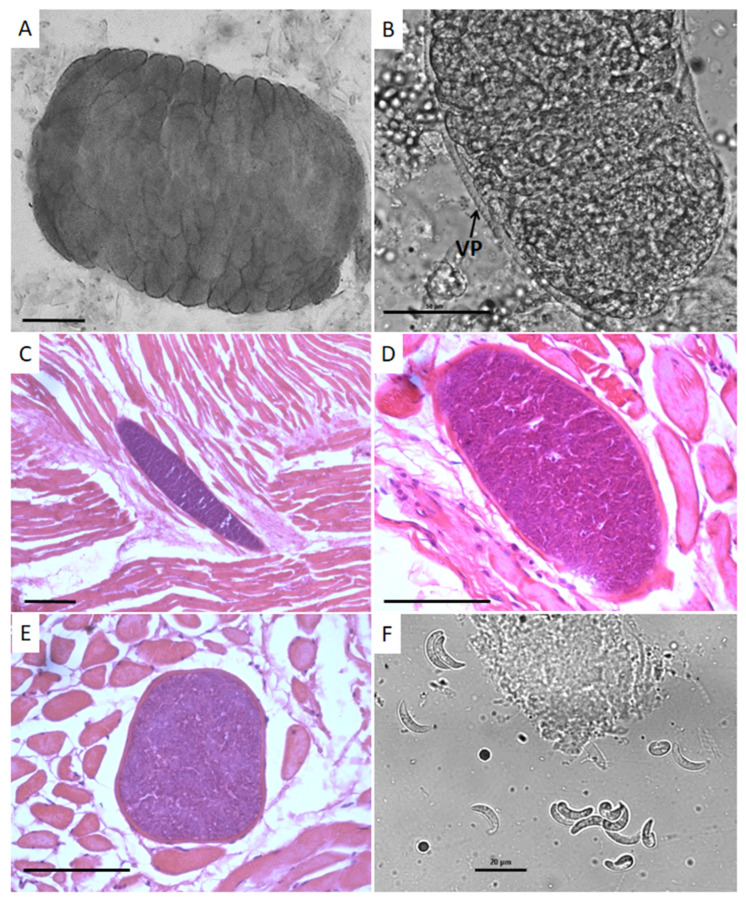
*Sarcocystis bertrami* in horse tissues observed using light microscopy. (**A**) A highly septated unstained sarcocyst from the tongue with numerous septa. Bar = 100 µm; (**B**) part of an unstained tissue cyst from the tongue with visible villar protrusions. Bar = 50 µm; (**C**) an HE-stained section from the tongue containing an elongated tissue cyst. Bar = 200 µm; (**D**) an elliptical tissue cyst in a tongue section stained by HE. Bar = 100 µm; (**E**) a globular tissue cyst from the tongue stained by HE. Bar = 100 µm; (**F**) free bradyzoites obtained by acid pepsin digestion from a fragment of gluteal muscle. Bar = 20 µm.

**Figure 2 animals-12-03491-f002:**
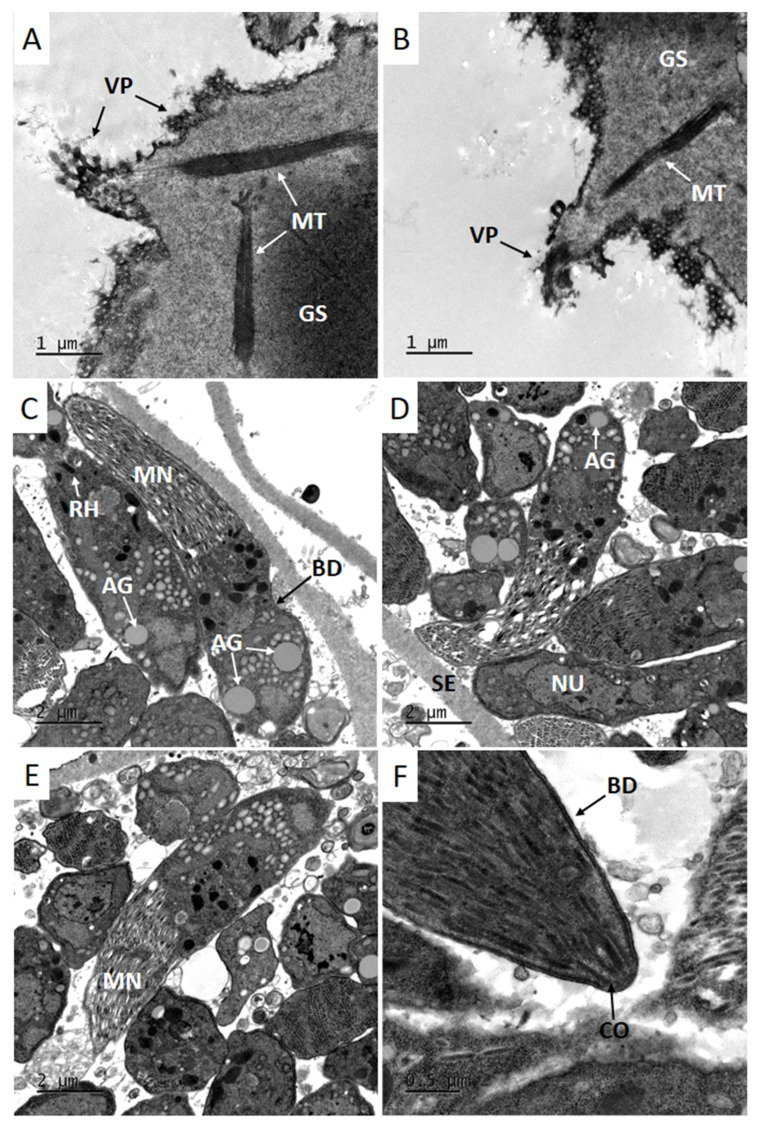
*Sarcocystis bertrami* from Brazilian horses observed using transmission electron microscopy of tongue tissues. (**A**,**B**) Villar protrusions (VP) on cyst walls and microtubules (MT) along the villi as well as in the granular substance; (**C**–**E**) bradyzoites observed at different angles containing numerous micronemes (MN), a moderate number of visible rhoptrias (RH), a single nucleus (NU) and amylopectin granules (AG); (**F**) the conoid (CO), which is located in the apical region of a bradyzoite.

## Data Availability

The data presented in this study are available on request from the corresponding author.

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
