# Peer review of "Brazilian Horses from Bahia State Are Highly Infected with Sarcocystis bertrami"

_animals, 2022, doi:10.3390/ani12243491_

Round 1
Reviewer 1 Report
The definitive host of Sarcocystis bertrami is the dog, and there is no evidence of pathological lesions in humans. This information needs to be removed from the abstract.
Author Response
Thank you for the time you dedicated to review our manuscript. We have addressed all questions you have raised and we believe the quality of the manuscript has been improved. Please, see below our response (R) to each question.
The definitive host of Sarcocystis bertrami is the dog, and there is no evidence of pathological lesions in humans. This information needs to be removed from the abstract.
R: That is right. Although suspected, there is no evidence of lesions in humans caused by ingestion of meat containing S. bertrami. This information was removed from the abstract.
The Introduction can be improved
R: Additional information was added in the second and third paragraphs of the introduction. Second paragraph: “however, lesions in humans caused by ingestion of horse meat containing Sarcocystis sp. has never been reported”. Third paragraph: “Species names of Sarcocystis spp. found in meat from equids have been debated in several studies [15-19]. Sarcocystis bertrami and Sarcocystis fayeri have been described in muscle tissues from donkeys, mules and horses [4-11]. The original classification of S. fayeri as a new species was mostly based on the prepatent period of sporocyst shedding by dogs and on the dimensions of excreted sporocysts [5]. Almost a decade after classification of S. fayeri, a study published in 1986 on Sarcosystis spp. derived from mules, donkeys and horses demonstrated that morphological differences of sarcocysts derived from these three animal species were so insignificant that all parasites seem to represent a single species [11]. In recent years, studies using morphological and genetic approaches have shown that S. bertrami and S. fayeri are the same species [17-19]. In the current work, the species name S. bertrami will be used to represent both S. bertrami and S. fayeri, and the latter will be referred as a synonymous of S. bertrami.”
Reviewer 2 Report
This manuscript reported the prevalence of sarcocysts of S. bertrmi in Brazilian hosres using the morphological and molecular methods. It provided new information concerning the distribution of the parasite. The results is reliable, herein it deserves to open in the journal.
Minor questions
Line 30, keyweords should include S. bertrami, Prevalence, Morphologic characteristics, Cox1
Figure 1B Based on our experiences, the image is impossible to be amplified 1000X, please check it.
Line 201: Here needs to supplement the accession number in GenBank, other than submission ID.
Author Response
Thank you for your accurate review. Please, see below our response (R) to each question.
Line 30, keywords should include S. bertrami, Prevalence, Morphologic characteristics, Cox1
R: We have added the suggested keywords. We have included the keyword “morphologic characteristic” in the singular form, as it seems to retrieve more data than the plural form.
Figure 1B Based on our experiences, the image is impossible to be amplified 1000X, please check it.
R: The correct magnification is 600X. The magnification of 1000X was substituted by 600X.
Line 201: Here needs to supplement the accession number in GenBank, other than submission ID.
R: The definitive accession numbers for the cox1 sequences were released by GenBank and have been added to the text (OP887040- OP887043).
Reviewer 3 Report
I'm afraid the manuscript has a limited revelance for international readers. In fact, it proposes no new methodologies and finds more or less what previous reports have shown in Europe, Asia, Africa...a high prevalence of S. bertrami. The same research group has previously detected high prevalences of the Sarcocystis genus in Brazil, even if not exactly the same species, and included a phrase of the possible S. bertrami presence in horses (Borges-Silva et al., 2020) : We concluded that Brazilian horses are exposed to distinct Sarcocystis species that generate different serological responses in exposed animals. Antigens in the range of 16 and 30 kDa are probably homologous in the two parasites. Exposure of the tested horses to other Sarcocystis species, such as Sarcocystis lindsayi, Sarcocystis speeri, and Sarcocystis fayeri, or Sarcocystis bertrami cannot be excluded in the current study.
I would only like to add a comment about lines 241-246: you are formulating and hypothesis based up on the fact that dogs are the definitive host of S. bertrami. However, you have not tested this hypothesis, and that must be clear for readers.
Author Response
Thank you for the time you dedicated to review our manuscript. Please, see below our response (R) to each question.
I would only like to add a comment about lines 241-246: you are formulating and hypothesis based up on the fact that dogs are the definitive host of S. bertrami. However, you have not tested this hypothesis, and that must be clear for readers.
R: In order to clarify our statement, we have included the following sentence in the discussion: “although Brazilian domestic dogs have also not been tested as definitive hosts of S. bertrami in Brazil”.
The introduction must be improved
R: Additional information was added in the second and third paragraphs of the introduction. Second paragraph: “however, lesions in humans caused by ingestion of horse meat containing Sarcocystis sp. has never been reported”. Third paragraph: “Species names of Sarcocystis spp. found in meat from equids have been debated in several studies [15-19]. Sarcocystis bertrami and Sarcocystis fayeri have been described in muscle tissues from donkeys, mules and horses [4-11]. The original classification of S. fayeri as a new species was mostly based on the prepatent period of sporocyst shedding by dogs and on the dimensions of excreted sporocysts [5]. Almost a decade after classification of S. fayeri, a study published in 1986 on Sarcosystis spp. derived from mules, donkeys and horses demonstrated that morphological differences of sarcocysts derived from these three animal species were so insignificant that all parasites seem to represent a single species [11]. In recent years, studies using morphological and genetic approaches have shown that S. bertrami and S. fayeri are the same species [17-19]. In the current work, the species name S. bertrami will be used to represent both S. bertrami and S. fayeri, and the latter will be referred as a synonymous of S. bertrami.”
Round 2
Reviewer 3 Report
The authors have added the information asked in previous report by myself and by the other referee. My main concern is about the originality of the report, since, as stated in the manuscript, similar results are obtained all over the world, and we must accept that equids are heavily infested by this parasite.
It's up to the Revue to decide wether this manuscript is sufficiently original to deserve publication as it is, or it must be summarised to a shorter report.
As a referee, as it refers to procedures, data etc, I have nothing else to comment. Is this equivalent to Accept in present form? That is the question. From my point of view, I can only say everything is correct and therefore, I'll mark "Accept in present form"